# Genetic Biomonitoring and Biodiversity Assessment Using Portable Sequencing Technologies: Current Uses and Future Directions

**DOI:** 10.3390/genes10110858

**Published:** 2019-10-29

**Authors:** Henrik Krehenwinkel, Aaron Pomerantz, Stefan Prost

**Affiliations:** 1Department of Biogeography, University of Trier, 54296 Trier, Germany; krehenwinkel@uni-trier.de; 2Department of Integrative Biology, University of California, Berkeley, CA-94720, USA; pomerantz_aaron@berkeley.edu; 3Marine Biology Laboratory, Woods Hole, MA-02543, USA; 4LOEWE-Centre for Translational Biodiversity Genomics, Senckenberg Museum, 60325 Frankfurt, Germany; 5South African National Biodiversity Institute, National Zoological Garden, Pretoria 0002, South Africa

**Keywords:** portable sequencing, nanopore sequencing, biodiversity monitoring, local capacity building, MinION, DNA barcoding, long read sequencing

## Abstract

We live in an era of unprecedented biodiversity loss, affecting the taxonomic composition of ecosystems worldwide. The immense task of quantifying human imprints on global ecosystems has been greatly simplified by developments in high-throughput DNA sequencing technology (HTS). Approaches like DNA metabarcoding enable the study of biological communities at unparalleled detail. However, current protocols for HTS-based biodiversity exploration have several drawbacks. They are usually based on short sequences, with limited taxonomic and phylogenetic information content. Access to expensive HTS technology is often restricted in developing countries. Ecosystems of particular conservation priority are often remote and hard to access, requiring extensive time from field collection to laboratory processing of specimens. The advent of inexpensive mobile laboratory and DNA sequencing technologies show great promise to facilitate monitoring projects in biodiversity hot-spots around the world. Recent attention has been given to portable DNA sequencing studies related to infectious organisms, such as bacteria and viruses, yet relatively few studies have focused on applying these tools to Eukaryotes, such as plants and animals. Here, we outline the current state of genetic biodiversity monitoring of higher Eukaryotes using Oxford Nanopore Technology’s MinION portable sequencing platform, as well as summarize areas of recent development.

## 1. Introduction

The past decades are distinguished by unprecedented global change, disrupting the integrity of ecosystems worldwide [1]. Mass extinctions of species are expected across the tree of life [2], with recent reports of large-scale declines in global insect populations receiving particular attention [3,4]. The detection of the human imprint on global biodiversity is critical for ecologists, conservation practitioners and policy makers to develop efficient countermeasures. This need has led to the emergence of numerous national and international biodiversity monitoring programs (e.g., [5]), which have to tackle the immense task of identifying millions of specimens in order to measure changes in ecosystems. 

Recent developments in high-throughput DNA sequencing (HTS) technologies have greatly facilitated the monitoring and characterization of biological communities. DNA barcoding [6], the assignment of taxonomic identities based on short amplicon sequences, has developed into an indispensable tool for genetic biomonitoring (Box 1, Figure 1). With ever decreasing sequencing cost and massive multiplexing capabilities of current HTS technologies, molecular barcodes can be generated for thousands of taxa in parallel [7]. Meta barcoding, the sequencing of barcodes from mixed community samples, allows the accurate detection of community compositions [8,9] and even interspecific interactions, like predation and parasitism [10,11,12] (Box 1, Figure 1). The approach can also be used to detect DNA traces left by species in their environment (e.g., soil or water), so-called environmental DNA (eDNA; [13,14]), and thus to passively characterize communities without the need for collecting actual specimens (Box 1).

Commonly applied amplicon sequencing technologies limit the fragment length of barcode sequences to a maximum of 500 bp when Illumina’s MiSeq or LifeSciences’ IonTorrent are used. For large-scale monitoring projects that utilize Illumina’s HiSeq or NovaSeq technologies, even shorter “mini-barcodes” must be used [15]. While short barcode sequences often contain sufficient variation to distinguish even closely related taxa, the phylogenetic resolution offered is limited. Phylogenetic diversity, however, is an essential component of biodiversity [16,17].

A popular approach to study whole communities without being limited to short fragment lengths is metagenomics, the generation of genomic reads for pooled individuals or community samples [18] (Box 1, Figure 1). These reads can be processed and compared to a database, or used to assemble (partial) genomes of different species present in the sample. Often, only a subset of the reads in the genomic library is utilized, for example to assemble multi-copy regions, such as mitochondrial or chloroplast DNA. This approach is referred to as genome skimming [19] (Box 1). The resulting long assembled sequences allow for DNA barcode analysis with high phylogenetic resolution [20]. Moreover, the direct sequencing of genomic DNA avoids the amplification bias commonly associated with PCR [21] and thus allows for more accurate quantitative recovery of taxa from a community sample [22]. However, compared to amplicon sequencing, metagenomics requires more complex and expensive library preparations. Additionally, only a small subset of DNA sequences is often retained for analysis, which requires a very high sequencing effort. This makes the method prohibitively expensive when large community samples have to be analyzed or for projects with limited funding. 

Technological developments in long-read sequencing now make it possible to sequence long PCR amplicons. So-called third generation sequencing platforms such as Pacific Biosciences’ Sequel have successfully been used to generate long-read DNA barcode data [23,24], with considerable phylogenetic resolution. At the same time, targeting particular amplicons in the genome makes it possible to minimize sequencing effort compared to genome skimming applications. With appropriate error correction applied, the method may also be suitable to generate long-read metabarcoding data [25,26]. 

Despite these advantages, and promising technological developments, current HTS based DNA barcoding approaches have two major drawbacks:(1)Limited accessibility of HTS: A significant disadvantage is the limited accessibility and high cost of most high-throughput DNA sequencers, which usually amount to > $100,000. Moreover, HTS often requires sophisticated laboratories to carry out library preparation and sequencing. Many labs simply do not have the budget to set up these systems. Scientists in developing countries with limited research infrastructure are most affected by this issue. However, developing countries in particular harbor a vast proportion of the world’s biodiversity and are therefore critical participants in the effort to measure anthropogenic impacts on the environment.(2)Long turnaround time: Ecosystems of particular conservation importance are often remote and not easily accessible. Samples have to be acquired in long expeditions, and processed in laboratories, sometimes days or even weeks away from sampling locations. International shipping of samples can require special permits for threatened and endangered species (regulated by CITES, https://cites.org), which can severely delay monitoring and conservation projects. From the beginning of a field expedition to the generation of genetic data, months or even years can pass. Often, local diversity at focal sites is declining rapidly, e.g., due to disease outbreaks and natural resource extraction, making accelerated assessments crucial.

Considering these drawbacks, biomonitoring approaches are required which are minimalistic in terms of logistics, price and accessibility. Ideally, they should enable researchers to generate barcode sequence data for large community samples in the field, without the need for transporting specimens to a laboratory or international shipping of samples. 

A minimalistic and highly portable approach for species monitoring was recently suggested based on quantitative PCR (qPCR). Compact, lightweight real-time PCR devices, such as the Biomeme qPCR machine, with shelf-stable reagents can be used to detect species in the field [27,28,29]. However, this approach relies on species-specific assays, and its taxonomic breadth is therefore very limited. This restricts its applicability to studies on the presence and absence of few target taxa and does not allow monitoring of communities. Additionally, the studied taxa have to be known a priori for this approach, as lineage specific primers have to be designed. Last, qPCR only detects the presence of a species, and does not provide phylogenetically informative sequence information. 

A highly promising alternative solution for portable biodiversity monitoring is offered by nanopore-based sequencing technologies (Oxford Nanopore Technologies). Nanopore sequencing is readily accessible to researchers, while at the same time can generate very long DNA sequences with high phylogenetic and taxonomic resolution. Thus, it is not surprising that past years have seen an increased application of nanopore sequencing for biodiversity explorations, with researchers from around the world capitalizing on its simplicity, accessibility, cost effectiveness and mobility. Here, we provide an overview of nanopore sequencing for biodiversity assessments. We specifically focus on ONT’s portable MinION sequencing platform, which can be used to carry out genetic biomonitoring in the field or in countries or areas with limited research infrastructure or funding. In this review, we outline the available technology and present an overview of available protocols and data analysis pipelines for portable biodiversity explorations. Lastly, we discuss advantages and disadvantages of the current technologies and approaches, and present an overview of future developments in the field.

Box 1Glossary of techniques commonly applied in genetic biomonitoring projects.**DNA barcoding:** Amplification and sequencing of short DNA fragments that contain diagnostic sequences to distinguish taxa. Species identifications are then carried out by comparisons to reference databases.**Metabarcoding:** Sequencing of mixed DNA barcode amplicons from bulk community or pooled samples. The qualitative taxon composition of the bulk sample can then be characterized based on the recovered barcode reads by comparisons against a reference database. PCR amplification bias can, however, skew quantitative inferences of taxon abundances in the community.**Metagenomics:** Shotgun sequencing of genomic DNA of a target taxon or bulk community. Longer contiguous sequences (particularly for multi-copy loci) can be generated by assembling the resulting reads, for example for whole mitochondrial genomes. By omitting PCR amplification, less biased quantitative assessments of communities are possible. The resulting long contigs also provide better phylogenetic resolution than short DNA barcodes alone.**Environmental (eDNA):** Environmental DNA traces left by organisms in their environment, e.g., plant pollen in soil or fish scales in water. eDNA can be enriched and sequenced by metabarcoding from e.g., water samples, allowing characterization of whole communities via passive sampling.**Genome Skimming:** Metagenomic approach in which only multi-copy loci (usually chloroplast or mitochondrial genomes) are retained. Contiguous sequences of these regions are then assembled and used for phylogenetics and community analysis.

## 2. Nanopore Sequencing for Portable Biodiversity Monitoring 

Recent years have seen an increased use of a small nanopore-based DNA sequencing platform, called the MinION (Oxford Nanopore Technologies, ONT). This device offers many characteristics suitable for genomic biomonitoring, as it is small in size, lightweight, inexpensive and allows for sequencing of several gigabases of DNA on a single flow cell. The device is capable of sequencing ultra-long reads [30], with a current upper limit of over 2 megabases [31]. Its small size at 10 × 3.2 × 2 cm and 90 grams, and the fact that it can be powered via USB, make this device a valuable tool for portable sequencing projects. Given the relatively small up-front and running costs, it is furthermore very interesting to researchers without much research funding.

In this technology, DNA fragments are funneled from one side of a biological membrane to the other through biological nanopores using a motor-protein (see, e.g., [32]). Depending on the version, these pores include a single (R9.x) or a double (R10) reader head, in which the presence of DNA bases restrict the flow of ions and thus enables the detection of current changes. These current changes (also referred to as “squiggles”) are then converted into base sequences, a process called basecalling. The MinION reads in 5-6 base pairs (5-6mers) at a time [33], though most of the signal originates from the three most central bases [34]. This plays a major role in the error profile of this technology, namely the issue calling the correct number of bases in homopolymer stretches. The newly developed double reader head in the R10 pore should help to overcome this issue by increasing the number of bases that make up the majority of the signal from three to six [34]. However, despite a raw read-error rate of about 5–25% for ONT’s R9.4 flow cells [35,36], highly accurate consensus sequences for DNA barcodes can be produced from nanopore read data [37,38,39]. Over the years ONT released different sequencing platforms, which vary in throughput (such as the MinION, the GridION and the PromethION). For the purpose of this review, we will only summarize and discuss applications of the inexpensive, portable MinION device as it can be used for sequencing in the field. 

Library preparation and sequencing workflow for the MinION are straightforward. It is also well suited for amplicon sequencing, and simple indexing strategies can be used to multiplex hundreds to thousands of amplicon samples [38,39,40,41]. Similar to dual indices in Illumina sequencing [42], short indices can be incorporated as 5’-tails into PCR primers. Based on these indices, samples can be computationally separated post sequencing. To account for the high error rate of raw nanopore reads, indices have to be sufficiently long. Previous studies suggest a minimum of 12bp [40] or 15bp [39]. However, even longer indices of 20 to 30 bp may be advisable, and indeed ONT’s barcoding kit uses 24bp indices now. Demultiplexing software developed for Illumina short read sequencing does not allow too many mismatches and is thus not suitable to demultiplex noisy nanopore reads. Specifically tailored software solutions are available, allowing several randomly distributed mismatches in the index (see “Bioinformatic pipelines for genetic biomonitoring using the MinION platform“ section below). 

The throughput of the MinION corresponds to approximately 20 Gb maximum output for R9.4 and will likely be around the same or slightly lower for R10 flow cells. Highly accurate consensus sequences can be generated at a coverage of as few as 10 reads [38,39]. This means that thousands of amplicon samples can be processed on a single sequencing run (see, e.g., [41]). While this is advantageous for large-scale analysis of ecosystems or DNA barcoding consortia, many end-users do not require such high-throughput. As a consequence, a lower throughput alternative was provided with ONT’s flongle flow cell, which was specially developed for projects with lower throughput demands (producing an average of 1–2 Gb output). Furthermore, MinION flow cells can be washed and reused multiple times, making it possible to split the sequencer’s throughput into multiple runs. The use of different index sets can circumvent any between-run cross contamination caused by imperfect removal of fragments from previous runs. 

## 3. Applications, and Advantages and Disadvantages of (Mobile) Nanopore Sequencing for Genomic Biomonitoring

Recent years have seen tremendous progress in miniaturization, leading to ever smaller and cheaper biotechnological devices available [43]. Common tasks like centrifugation, gel electrophoresis, nucleic acid quantification, PCR, restriction digestion or ligation can now be performed routinely using portable and battery or even hand-powered devices [38,44,45]. 

The first step of isolating genetic material from samples typically entails lysing the tissue, which can be carried out by physically macerating the sample with a pestle or using handheld bead-beating instruments such as the Terralyzer (Zymo Research). Next, DNA extraction and purification steps can be conducted using a variety of strategies or commercial kits. Spin-column based nucleotide extraction kits have been shown to be useful for field experiments, as reagents can be stored at room temperature and the primary piece of equipment required can be a small commercial centrifuge or even a 3D-printed hand-powered centrifuge device [44]. Miniaturized thermocyclers, such as those manufactured by miniPCR or MiniOne Systems, can be powered by a portable battery back to carry out PCR amplification or heat-block steps in remote settings [38]. Finally, ONT sample library preparation is relatively straightforward and can be carried out in under two hours.

In combination with mobile laboratory equipment, nanopore sequencing can be used for field-based sequencing experiments even in very remote areas with limited or no access to electricity [38,46] (see Table 1 for applications). The advantage of the MinION’s portability was recognized early on with the applications of the technology to survey viral outbreaks [47], such as Ebola in Guinea [48] and Zika in Brazil [49]. Its potential for conservation and biodiversity monitoring was implemented shortly after [38,46,50]. In References [38] and [46], the authors showed the feasibility of in-the-field DNA barcoding for biodiversity assessments and local capacity building in areas with high biodiversity. This offers the potential to carry out genetic biomonitoring in remote areas of the world, guide sampling strategies in the field in real-time and carry out large-scale monitoring projects in areas with little research infrastructure [38]. In order to increase its utility even further, many studies present pipelines and automated approaches to process the sequencing data. These are summarized in the section “Bioinformatic pipelines for genetic biomonitoring using the MinION platform” below and Table 2.

The massive multiplexing capabilities of the MinION allow the generation of thousands of DNA barcodes in a single sequencing run (see, e.g. [41]). The price per barcode can further be reduced by direct PCR [51], which avoids DNA extractions and additional levels of sample indexing [52]. The resulting massive processing capability led to the suggestion to use DNA barcoding of whole community samples to assist taxonomic discoveries. Usually collections of specimens are presorted morphologically and DNA barcodes generated only for target groups. This is a very labor intensive and time-consuming task, often hindering and delaying larger monitoring projects. In order to reduce the need for morphological sorting, a reverse workflow can be applied [41,53]. In this workflow all specimens collected at a site are DNA barcoded, and the resulting barcode information used to identify groups that warrant further morphological investigation and treatment. So far, the largest study sorted and DNA barcoded over 7000 phorid flies, sequencing over 4500 individuals on a single MinION flow cell [41]. Even though the price per DNA barcode has rapidly dropped in recent years, presorting and single specimen processing can require extensive time for large ecosystem-wide collection of specimens substantially increasing project costs indirectly [54]. 

A more efficient large-scale monitoring strategy would include metabarcoding of thousands of unsorted specimens. This would be particularly helpful for field-based applications. However, error rates of the individual reads (5–25% [35,36]) currently limit the MinION’s application for metabarcoding. A clear distinction between error and biological sequence could only be achieved, when the genetic distance between members of the sequenced community is higher than the read-error rate. However, this assumption is rarely met in real communities. The high error rate also limits the possibility for biologically meaningful operational taxonomic unit (OTU) clustering. In OTU clustering reads are grouped based on a predefined percent similarity. However, high error rates can lead to problems clustering reads even from one and the same individual. This can be mitigated to a certain extent by relying on long amplicons and comparing them to a reference database. As read errors, apart from homopolymer errors, are mostly randomly clustered along the sequence, a well-developed reference database will help assigning reads to certain taxonomic level. A better, though not portable, alternative for metabarcoding may be offered by Pacific Biosciences. The circular sequencing offered by Pacific Biosciences’ Sequel considerably reduces the raw read error, making this a promising technology for metabarcoding applications [25,26]. A similar technique is not currently available for nanopore sequencing. However, the application of the two-sensor R10 pore, and the associated drop in error rate, will likely increase the MinION’s use for metabarcoding (see, e.g., [55]). Furthermore, Calus and colleagues recently showed the huge potential for rolling-circle based amplification for metabarcoding applications on the MinION [37]. In this technique individual read-error rates are mitigated by sequencing long stretches of multiple replicates of the same DNA barcode. Requiring only a minimum of three concatemers to remove all errors but deletions in homopolymer regions from the barcodes makes this technique very interesting for large-scale metabarcoding projects [37]. While ONT or Pacific Biosciences platforms offer great potential for large-scale metabarcoding projects, only ONT’s MinION platform is small and inexpensive enough to be used in developing countries or remote areas. Indeed, the great potential of nanopore sequencing for metabarcoding in remote areas was pointed out early on in Reference [56]. 

A major advantage of the MinION platform beside its scalability via multiplexing is the long read length compared to other HTS platforms (such as Illumina’s or BGI’s sequencing platforms). While short DNA barcodes are often sufficient for delineation of species, they do not offer much phylogenetic resolution. Phylogenetic diversity, however, is an important metric for community analysis [16,17]. Resolving the phylogenetic structure within a community is also essential to explore patterns of trait evolution [57]. Third generation sequencing platforms (such as Pacific Biosciences or ONT) offer the unique opportunity to sequence long DNA barcodes that both offer power to delineate species and also provide phylogenetic resolution [23,25]. A popular region for this purpose is the nuclear ribosomal DNA (rDNA) cluster. While conserved gene regions of rDNA can be used to design highly universal primers and resolve old divergences [58], the fast-evolving internal transcribed spacers (ITS) can be used to resolve relations of closely related sister species [59]. Recently, Krehenwinkel and colleagues showed that long rDNA sequencing on the MinION platform also offers great potential for in-the-field applications [39]. A drawback of the rDNA approach is that the amplicon length can differ significantly between taxa, especially due to high length polymorphism of the ITS regions [60]. Length differences can lead to biases during amplification, which can strongly bias metabarcoding applications [39]. For long-read animal barcoding, long mitochondrial amplicons, which include the COI gene, may also be particularly attractive, considering the available well-developed COI DNA barcode reference databases [6].

The long-read length of nanopore sequencing also offers great opportunities for pooled sample or bulk community metagenomics [50,61]. A major drawback of this technique is the current lack of whole-genome reference data for many species [54]. Often, only traditional DNA barcode regions, such as COI for animals [6] or ITS for fungi [59], are well represented in databases. Consequently, the vast majority of metagenomic reads cannot be assigned to taxon level and have to be excluded from the analysis. Recent work suggests a solution to this problem by “reverse metagenomics” [61]. Here, a whole-genome reference database of low coverage Illumina short reads is generated for target taxa in the community. Using nanopore long-read sequencing of bulk community samples, this then makes it possible to link the long nanopore genomic reads to separate taxa from the short-read library. This way, long genome-wide sequence data can be generated for whole community samples. 

## 4. Bioinformatic Pipelines for Genetic Biomonitoring Using the MinION Platform

In recent years, several pipelines have been presented to analyze DNA- and metabarcoding or metagenomic data produced by the MinION platform (see Table 2; see Figure 2 for a schematic of the general pipeline steps).

The first step in each pipeline is the basecalling, the conversion of the current pattern (“squiggles”) obtained from the MinION platform into base sequences. This is usually carried out using ONT’s in-house Albacore (https://github.com/Albacore/albacore), Flappie (https://github.com/nanoporetech/flappie) or Albacore’s successor Guppy (https://community.nanoporetech.com). Both Guppy and Flappie offer improved accuracy due to the newly implemented flip-flop model [64]. Independent software solutions such as Chiron [65], DeepNano [66] are available. A detailed comparison of nanopore basecalling software tools can be found in Reference [67] and on https://github.com/rrwick/Basecalling-comparison/tree/v5.1. 

In the next step the sequencing reads are usually demultiplexed. This can either be carried out using ONT’s basecalling software (Albacore or Guppy), specific in-house software solutions offered by ONT such as qcat (https://github.com/nanoporetech/qcat) or third-party applications such as porechop (https://github.com/rrwick/Porechop**)**, DeepBinner [36] or Minibar [39]. While ONT’s in-house solutions are easy to use, they usually offer little flexibility for different indexing schemes as they are developed for ONT’s in-house indexing kits. In contrast, third-party applications such as porechop, DeepBinner or Minibar offer much more flexibility as they can handle customized indexing. Furthermore, they allow for different filtering stringency, and in the case of Minibar also allow for further demultiplexing of the amplicons in a multi-locus pool (based on which PCR primers were used). Here, PCR primers can thus be used as additional indices, to demultiplex loci from multiplex PCRs.

Most pipelines include a quality filtering step either before or after the demultiplexing of the sequencing data. Here, reads are usually filtered based on quality scores (such as Phred scores) and read lengths. The latter is carried out to remove chimeras. Commonly used tools include NanoFilt (https://github.com/wdecoster/nanofilt) or seqtk (https://github.com/lh3/seqtk). 

After quality filtering the sequencing reads of each DNA barcode are assembled. This can be carried out using (1) de novo assembly tools such as canu [67] or allele wrangler (https://github.com/transplantation-immunology-maastricht/allele-wrangler), or (2) by clustering highly similar reads using, e.g., vsearch [68] or IsONclust [69] and subsequent alignment using, e.g., MAFFT [70] or LAST [71]. 

Most nanopore DNA barcoding pipelines include a post-assembly consensus sequence polishing step. Several software solutions are available, these include ONT’s Medaka (https://github.com/nanoporetech/medaka) and third-party software such as Nanopolish [72] or RACON [73]. In this step, the read data is mapped back against the consensus sequence and errors are corrected using information from multiple reads covering a certain base in the consensus and/or the raw current data stored in fast5 files (default output format of ONT’s sequencing platforms). In the last step, the final polished consensus sequences are then compared against reference databases such as the Barcode of Life Data System (BOLD; for COI barcodes; [74]) or NCBI’s GenBank [75].

For metabarcoding applications, filtered reads are usually clustered according to their similarity into OTUs using tools such as vsearch [68]. Individual OTUs are compared against a reference database. Many pipelines, such as MINDS [76] or WIMP [77] are based on the taxonomic classification tool Centrifuge [78].

## 5. Mobile Sequencing as A Tool for Local Capacity Building and Education

Mobile laboratory equipment offers great opportunities for local capacity building and education, for several reasons: mobile laboratory technologies are often designed to be durable, inexpensive and largely independent from electricity, Internet access or a cold chain. These factors can enable researchers in low resourced communities, who often lack access to sufficient funding or infrastructure [38,82]. Until recently, many developing countries or local researchers were dependent on international collaborations to carry out conservation genetic or genetic biodiversity monitoring, cultivating a so-called “conservation colonialism”. The advent of portable technologies offers great possibilities to change the current system [38], and indeed more and more community-based or local capacity development projects are starting to arise (see, e.g., [62,83]).

A crucial part of strengthening local capacity is science education. Similar to research, mobile technologies offer great promise in this area. Genetic sequencing has been used in classroom environments as an effective teaching tool [84,85], as well as in-the-field [62,83]. The latter case further offers great opportunities for local capacity building via interactions between local and international students. In addition to permitting scientists to conduct real-time research in the field, these tools provide exceptional teaching tools to empower students.

## 6. Outlook

Mobile sequencing technologies are rapidly evolving as sequencing devices get smaller and cheaper. Ever decreasing error rates promise great potential of mobile technologies for genetic biomonitoring even in countries with limited research infrastructure and funding. Beside strengthening research in less developed countries, often harboring most of the world’s biodiversity, mobile technologies also avoid the need for export and import permits for specimens, which can substantially delay biodiversity assessments. It is important to note that collection permits are usually still needed, and some countries even require special permits to carry out genetic research on specimens collected within the country. While mobile sequencing technologies may alleviate the bureaucratic obstacles to field-based biodiversity explorations, this may also lead to the danger of increased misuse and unsanctioned genetic research. 

Beside biodiversity monitoring, the MinION platform is interesting for other conservation and biodiversity related fields, such as wildlife forensics [86]. However, due to its high error rate in homogeneous sequences, the current technology does not allow for reliable sequencing of microsatellites (STRs), a standard marker in forensics [87]. Techniques such as genome skimming using the MinION device have been shown to work well for species identifications [63]. However, their application in wildlife forensics is currently impeded by a lack of studies validating genome skimming as a reliable molecular technique for legal case work or validation of the MinION platform itself. Even though portable sequencing equipment shows promise for screening at boarders or airports (for wildlife forensics or pest control), these techniques are still too funding and labor intensive to be regularly performed by boarder control or customs agents. Moreover, handling of mobile sequencing equipment still requires basic training in molecular biology, which is rarely available for law-enforcement personnel and compared to biodiversity monitoring the substantially lower throughput increases per sample expenses considerably.

Museum collections have gained importance for genetic biodiversity research in recent years [88]. Historic collections can, e.g., be compared to current biodiversity, or be used to generate reference databases. A prevalent issue for genomic studies based on museum collections is high DNA degradation [89,90]. A common solution for DNA- or metabarcoding applications based on degraded DNA is the use of ‘mini-barcodes’ (short amplicons). Unfortunately, many mini-barcodes are shorter than the minimum read length (~200bp) required by ONT platforms. Recently, Wilson and colleagues showed that rolling-circle PCR can be used to (a) overcome this minimum length requirement, while at the same time (2) generating highly accurate mini-barcode reads [91]. This opens up the possibility to utilize even heavily degraded DNA from museum specimens for MinION-based biomonitoring projects. However, inclusion of a rolling-circle step increases the costs per sample substantially and might not be efficiently performed in remote locations or when sophisticated laboratories are missing. 

Another interesting implementation for genetic biomonitoring, especially the monitoring of pest species or diseases, is the molecule-by-molecule real-time selective sequencing or “Read Until” developed for the MinION platform [92]. This technique enables the selective sequencing of certain target species. Squiggle patterns are analyzed in real-time and compared to a reference. If the DNA fragment matches the target species, the read will be sequenced while off-target DNA will be rejected by reversing the pore bias to eject the strand [92].

Finally, although many studies have shown the feasibility of MinION-based biomonitoring in recent years, most of these serve as proofs of concept. Large-scale practical applications of the proposed methodology are still mostly missing. Hopefully, future research will close this gap and contribute to establishing portable biodiversity monitoring as an integral part of future biodiversity genomics. 

## 7. Conclusions

Developments in high-throughput sequencing technology enabled a great leap forward for biodiversity research. Biodiversity genomics enables researchers to characterize whole species communities with great taxonomic and phylogenetic resolution, with sample processing cost constantly decreasing. This in turn provides researchers with an essential toolkit for the tremendous task to monitor global ecosystems and their responses to global change. The increasing read length of third generation sequencing technologies greatly contributed to this development. Mobile sequencing technology fills its own niche in the field of genetic biodiversity monitoring by providing (1) cost efficient and accessible solutions for biodiversity assessment to researchers around the globe and (2) by opening up the possibility to directly generate sequence information in the field. This is particularly important for biodiversity research in remote areas or when research infrastructure is missing, and especially when time is at the essence, e.g., from human-induced habitat destruction or during an outbreak of a wildlife disease (such as the chytrid fungus in amphibians [93]). 

## Figures and Tables

**Figure 1 genes-10-00858-f001:**
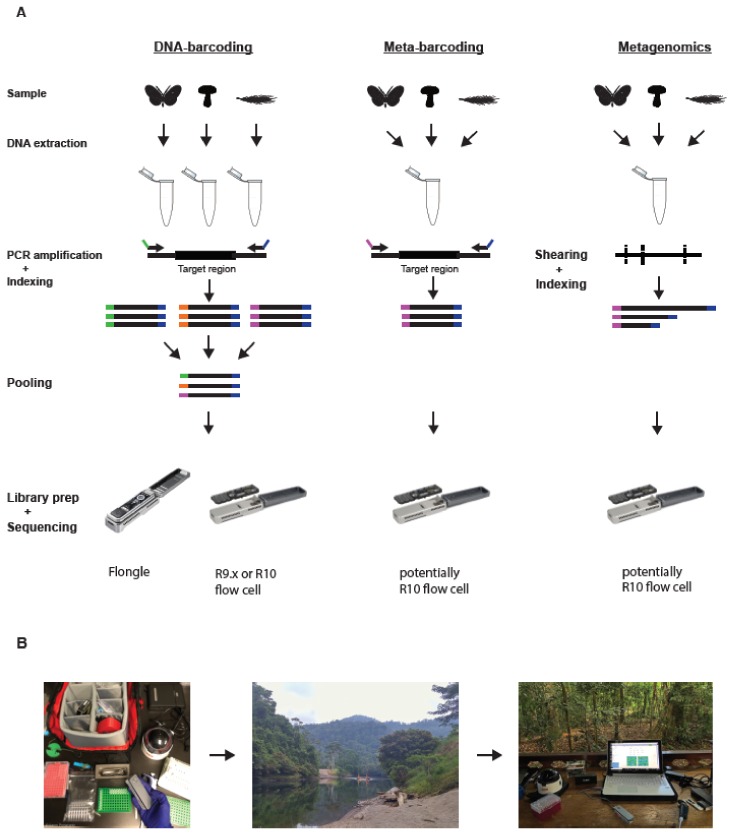
(**A**) Schematic of different genetic biomonitoring approaches using the MinION platform. (**B**) Schematic of the application of the mobile sequencing laboratory. (**A**) Left panel: DNA barcoding involves extracting DNA and performing PCR amplification from individual specimens. Amplicons are tagged with unique dual indices and then pooled into a single library and sequenced. Middle panel: Metabarcoding involves DNA extraction and PCR amplification from a bulk community sample. Each individual metabarcoding library can be dual indexed to enable pooling of different bulk samples. Right panel: Metagenomics involves extracting DNA directly from a bulk environmental sample. The DNA is then sheared to small fragment sizes and an indexed library constructed, before sequencing. Due to its long read length, shearing is not necessary for MinION-based metagenomics. The application of the MinION platform for metabarcoding and metagenomics is currently restricted due to the high error rates of R9.x flow cells. Changes to the pores (discussed above) in R10 flow cells are supposed to lower its error rates considerably, potentially enabling the use of MinION for these applications. (**B**) Left photo: A typical portable laboratory or “lab in a backpack” setup. Middle photo: Due to its limited size and cost, the portable laboratory can be transported even to remote field locations. Right photo: The portable laboratory setup in the jungle of Panama.

**Figure 2 genes-10-00858-f002:**
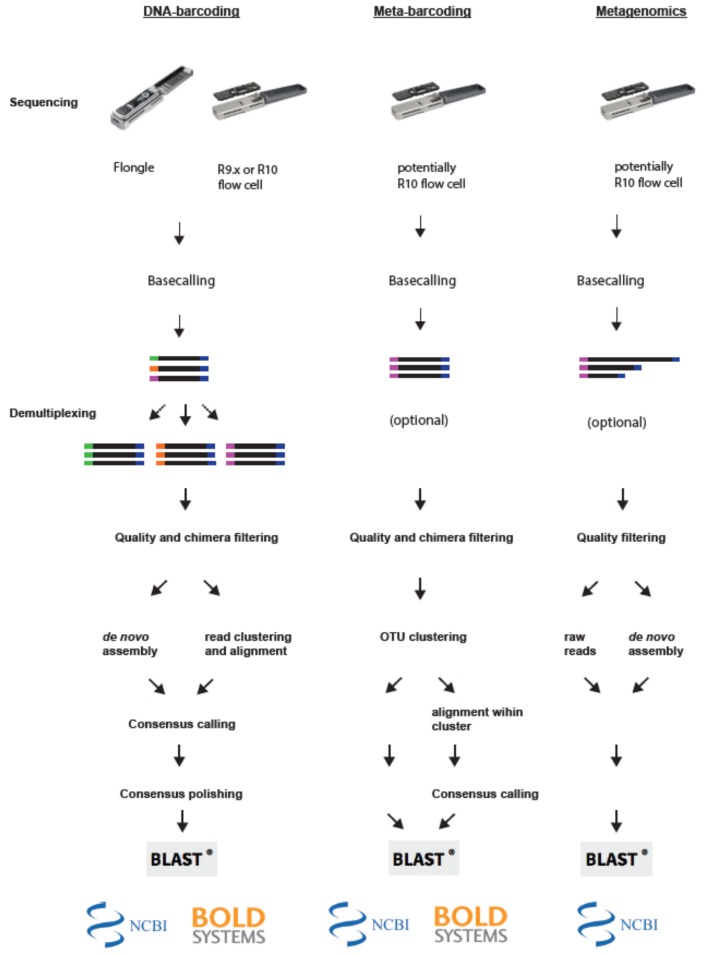
Principle steps of bioinformatic pipelines for MinION-based biodiversity assessment approaches. Left panel: For DNA barcoding the reads are first base-called and demultiplexed. Next reads are filtered and depending on the applied pipeline either assembled de novo, or clustered and subsequently aligned. Lastly, the consensus sequences are polished and blasted against a database. Middle panel: For Metabarcoding applications the reads are first base-called and quality filtered. Next reads are clustered into OTUs and then blasted against a database directly or aligned to create individual consensus sequences. Right panel: For metagenomic applications reads are first base-called and filtered. Subsequently, reads are either blasted against a database directly, or assembled de novo into contigs and subsequently blasted against a database.

**Table 1 genes-10-00858-t001:** Applications of the MinION sequencing platform for biomonitoring of higher eukaryote biodiversity.

	Citation	Purpose
**DNA barcoding**	[46]	To test the feasibility of in-the-field DNA barcoding. The field-application was carried out in Tanzania.
	[38]	To test the feasibility of in-the-field DNA barcoding. The field-application was carried out in Ecuador.
	[39]	To investigate the feasibility of using long-read DNA barcodes for biodiversity monitoring. The field-application was carried out in Peru.
	[62]	Proof of concept study of in-the-field DNA barcoding. The field-application was carried out in Madagascar.
	[41]	To document Phoridae (Diptera) biodiversity in the Kibale National Park in Uganda.
**Metabarcoding**	[39]	To test the usability of long-read rDNA barcodes for metabarcoding applications. The field-application was carried out in Peru.
**Metagenomics**	[61]	To test the feasibility of reverse metagenomics for species identification using the MinION platform.
**Genome skimming**	[50]	To test the feasibility of genome skimming for species identification using the MinION platform. The field-application was carried out in Wales.
	[63]	To test the feasibility of genome skimming using the MinION platform for species identification of highly-traded shark species.

**Table 2 genes-10-00858-t002:** Available pipelines for bioinformatic processing of genetic monitoring data generated with the MinION platform.

Name	Genetic Monitoring Technique	Programs Used in the Pipeline	Citation
	DNA barcoding	Based on de novo assembly using canu [67] and polishing using Nanopolish [72]	[38]
	DNA barcoding	Based on de novo assembly using Allele Wrangler (https://github.com/transplantation-immunology/allele-wrangler/) and polishing using racon [73]	[39]
	DNA barcoding	Based on alignments with MAFFT [70] and polishing using racon [73]	[40]
ONTrack	DNA barcoding	Based on clustering using vsearch [68], alignment with MAFFT [70] and subsequent polishing using Nanopolish [72]	[79]
NanoAmpli-Seq	Metabarcoding	Extension of the intramolecular-ligated nanopore consensus sequencing (INC-Seq) protocol [80]	[37]
What’s in my pot? (WIMP)	Metabarcoding	Stand-alone tool	[77]
MinION Detection Software (MINDS)	Metagenomics	Based on the based on the Centrifuge classification engine [78]	[76]
Nanopipe	Can be used for metagenomics	Based on LAST alignments [71]	[81]

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
