# Peer review of "Genetic Biomonitoring and Biodiversity Assessment Using Portable Sequencing Technologies: Current Uses and Future Directions"

_genes, 2019, doi:10.3390/genes10110858_

Round 1
Reviewer 1 Report
The premise is good and the authors provide a useful overview of the current state of MinION use. The introduction and background were well written. Sample preparation (from source to purified DNA and/or library prep) was minimally discussed, particularly in the context of mobile sequencing. The paper would benefit from discussion of detection limits and the relationship of those limits to the application space. The paper would further benefit from a description of what is needed to enable nascent applications. Reviews typically attempt to be comprehensive, thus, I would strongly encourage the authors to synthesize a more comprehensive view by addressing the above mentioned areas.
Detailed comments:
Lines 101-113: might be reasonable to cite one or more key nanopore studies
Line 122: “does not need an external power supply” -> perhaps more accurate is “can be powered via USB”
Line 126: “so-called nanopores” -> suggest “biological nanopores”
Line 127: The text describes one or two “sensors” -> the nanopores themselves do not have sensors. They have narrow regions in which the presence of DNA bases restricts the flow of ions. For the R10 pores, there are two of these regions, leading to, I imagine, more unique ionic current signatures. I suggest sticking “one (R9.x) or two regions of the pore in which the presence of DNA bases restricts the flow of ions”
Line 128-129: The speed is not at all constant, and varies by several orders of magnitude, depending upon the molecular motion of the motor protein as it unzips DNA. However, the average speed can be around 400 bp/s. The accuracy comes from the ability to make multiple ionic current measurements during each ratchet step, which provides the information required for any basecalling algorithm, but does not insure accurate basecalling (necessary, but not sufficient). Please adjust the text accordingly.
Line 130-132: Here you discuss how the ionic current signal originates from 5-6 bases. This may be a good spot to mention the difference between R9.x and R10.
Line 152: R10 is still not well characterized in terms of throughput and is currently a bit lower than R9.x.
Figure 1B: it is not really clear what is being represented in the flow of images from the figure and figure caption alone.
Lines 250-254: Given that the article discusses flongle and R10, it seems reasonable to ensure the basecalling comments are also up to date. With version 19.05.0, MinKNOW incorporated Flip-flop basecaller with regular and high accuracy modes:
Table 2: It’s not clear what the blank “-“ entries mean. Would it be possible to add some description even if a tool name is not available?
Author Response
Please, find our comments in the attached word document.

Reviewer 2 Report
Dear Authors,
The manuscript no. genes-607483 “Genetic biomonitoring and biodiversity assessment using portable sequencing technologies: current uses and future directions” is an overview, which provides description and comparison of different sequencing technologies with nanopore sequencing, particularly the ONT’s portable MinION sequencing platform, for biodiversity analysis. For this reason, this work can be considered as essential and valuable. My minor comments are listed below:
Page 1, title: In the manuscript, the Authors focused on the application, pros and cons of MinION platform. In my opinion, the title is too general and should be modify. Moreover, please remove the dot from the title line.
Page 1, abstract: Also in the abstract, the Authors should indicate on which platform they focus in the manuscript.
Page 10, line 309: Please modify the sentence: “Technologies get smaller and cheaper”
Maybe ‘devices’ instead of ‘technologies’ will be more precisely.
Table 1-2: Please, change the order of the columns in tables. Citations should be the last.
Please check the spaces throughout the article.
Author Response

(The authors gave the same response as above.)
